# The Colombian Zika Virus Isolate (COL345Si) Replicates in Prostate Adenocarcinoma Cells and Modulates the Antiviral Response

**DOI:** 10.3390/microorganisms10122420

**Published:** 2022-12-07

**Authors:** Yaneth Miranda Brand, Astrid M. Bedoya, Liliana Betancur-Galvis, Juan Carlos Gallego-Gómez

**Affiliations:** 1Grupo de Investigación Dermatológica, Universidad de Antioquia, Medellín 050010, Colombia; 2Grupo de Medicina Molecular y de Translación, Universidad de Antioquia, Medellín 050010, Colombia; 3Grupo Microbiología Ambiental, Escuela de Microbiología, Universidad de Antioquia, Medellín 050010, Colombia

**Keywords:** ZIKV, flavivirus, prostate, permissive infection, antiviral response

## Abstract

Zika virus (ZIKV), a flavivirus that is mainly transmitted by *A. aegypti* and *A. albopictus* and sexual transmission, has been documented and described. The ZIKV RNA detection in the semen of vasectomized men indicates that accessory glands such as the prostate could be a site of virus replication. In this study, we characterized the ZIKV infection, evaluated the antiviral profile, and demonstrated the AXL and TIM-1 expression on the PC3 prostate cell line. It was also determined that PC3 cells are susceptible and permissive to ZIKV infection without altering the cell viability or causing a cytopathic effect. The antiviral profile suggests that the PC3 cells modulate the antiviral response through the suppressor molecule expression, SOCS-1, during a ZIKV infection.

## 1. Introduction

Zika virus (ZIKV) is a flavivirus transmitted by the *Aedes aegypti* and *A. albopictus* vectors [1]. However, the clinical findings indicate that it is also transmitted through sexual transmission from person to person [2]. Although ZIKV has been found in various body fluids (saliva, blood, and urine), there is evidence that indicates the persistence of the virus in semen [3], as RNA genome and infectious particles have been detected up to 69 and 304 days [4], respectively, after the onset of symptoms. Therefore, the male reproductive tract (MRT) has shown to be an efficient transmission site for the virus since the testes are immune-privileged sites where the virus can persist for a long period [4].

In vasectomized patients the presence of the virus has been verified through a serological analysis, an RNA genome detection, and viral isolation up to 69 and 77 days post-symptom onset [4]. It has been demonstrated that the RNA ZIKV persists in the male reproductive tissues of C57BL/6J mice [5] and New World monkey species (NWMS) [5,6]. The ZIKV infection in the prostate was observed until 21 and 45 days post-infection in NWMS and C57BL/6J mice, respectively [6].

Prostatic diseases are linked with fertility alterations [7] as the prostate plays an important role in male fertility, particularly in the activation of sperm maturation. The prostatic fluid is enriched with Zn^2+^, citrate, and kallikreins, among others this leads to the homeostasis of the prostate epithelium [7]. The presence of receptors of the TAM (Tyro3, AXL, and Mer) [8] and TIM family [3] on the prostate gland [9] are decisive for ZIKV infection, and they indicate the tropism of the ZIKV, and this could explain the virus isolated of vasectomized patients and the detection viral RNA in the prostate tissues of mice and monkeys.

On the other hand, during the course of the illness and self-limiting febrile symptoms in most of the flaviviruses (FVs) cases, there is a key role of the innate immunity-comprising type I interferons (IFN-α, β), type II interferon (IFN-γ), and type III interferons (IFN-λ1–4), nonetheless, it is known that FVs inhibit the interferon response through some viral proteins [10,11]. Likewise, the AXL receptor, in addition to conferring susceptibility to the infection, can activate immunosuppressive molecules that regulate the antiviral response [12]. During a ZIKV infection in Sertoli cells [13], the transcription factor STAT1, an important molecule in the AXL-IFNAR signaling pathway, is activated; promoting the expression of suppressor of cytokine signaling 1 (SOCS1) and the restriction of the interferon response [13]. The inhibition of AXL during a ZIKV infection in Sertoli cells causes: (i) the inactivation of STAT1, (ii) the reduction of SOCS-1, and (iii) an interferon-induced antiviral gene (ISG) expression that decreases the viral replication [13]. The immune evasion strategies of ZIKV could contribute to viral persistence in the prostate gland. Therefore, it is important to understand in detail the way ZIKV infects a cellular line, encompassing several findings, starting from the interpretation of the virus genome, viral replication, including growth dynamics, and the innate immune response once the infection has been established.

Due to the difficult maintenance of primary cell cultures for the evaluation of a ZIKV infection in the MRT, we employed immortalized prostate epithelial cells (PC3), a cell line with the functional pathway of the interferon response [14].

To investigate the susceptibility and permissiveness of the ZIKV infection in the human prostate, AXL and TIM-1 receptor detection was carried out, and the viral fitness was analyzed by several methodologies.

Our research revealed that PC3 cells are susceptible and permissive to a ZIKV infection through the control of the interferon response by SOCS-1, allowing uniquely balancing viral replication and cell survival in PC3 cells of up to 120 h.p.i. These findings indicate a possible persistence of a ZIKV infection in PC3 cells.

## 2. Materials and Methods

### 2.1. Cells and Virus

To perform the biological assays, the PC3 ATCC^®^ CRL1435 (prostate adenocarcinoma) cells and Vero E6 ATCC^®^ (kidney of an African green monkey) cells were used maintained according to the specifications of the American Type Culture Collection (ATCC).

Zika virus COL345Si GenBank: MH179341.1 passage # 6 was isolated from *Aedes aegypti*. For the viral amplification, the Vero E6 cell line was used. The cells were infected with MOI = 0.01, and the infected cells were incubated for three days until as observed a cytopathic effect; then, the supernatants were collected and stored at −80 °C until their use.

### 2.2. Characterization of the Colombian Isolate of the Zika Virus COL345Si (MH179341.1)

Four ZIKV consensus sequences encoded in the gene bank: MH179341.1, MH544701, KJ501215.1, and KU955593.1 were aligned in the program MUSCLE. Subsequently, phylogenetic trees based on the complete gene sequence were made, and their evolutionary history was inferred using the Maximum Likelihood (ML) method of the MEGA 6.0 software, with a bootstrap test (1000 replicates) [15]. According to the jModelTest tool, the best nucleotide substitution model that was adjusted to the sequence matrix considering the Akaike (AIC) and Bayesian (BIC) information was Tamura–Nei [15].

### 2.3. Detection of AXL and TIM-1 Receptors

The expression of the cell receptors was analyzed in 2 × 10^4^ cells in the presence or absence of the infection. For this, flow cytometry, In-Cell Western (ICW), and fluorescence microscopy were employed.

For the ICW method, the cell monolayer was fixed (PFA 4%), and permeabilized (triton X-100) labeling was performed with a 1:200 dilution of anti-AXL (ab89224) and anti-TIM-1 (Bio-legend, San Diego, CA, USA) antibodies according to the protocols described, and they were revealed with IRDye 680LT (LI-COR Biosciences, Lincoln, NE, USA). The anti-α/β-tubulin (TU-10, Invitrogen™, Waltham, MA, USA) was used as a housekeeping gene, and it was revealed with IRDye 800CW (LI-COR Biosciences). The detection was performed with Odyssey (LI-COR Biosciences). The results were normalized from three independent experiments.

For the receptor detection by flow cytometry and fluorescence microscopy, the cells were trypsinized and fixed with 4% PFA, respectively. Subsequently, the labeling was performed with a 1:200 dilution of anti-AXL (ab89224) and anti-TIM-1 (Biolegend 353902, San Diego, CA, USA) antibodies according to the protocols that are described. Alexa Fluor 488 secondary antibody (Goat anti-mouse and anti-rabbit IgG H (Invitrogen Life Technologies, San Diego, CA, USA) was used and the nuclei staining was performed with Hoechst33342 (H1399, Invitrogen Life Technologies, San Diego, CA, USA) [16]. Finally, the samples were analyzed using a cytometer (BD Facs Canto II, by Becton Dickinson, Franklin Lakes, NJ, USA) and an epifluorescence microscope (60× objective).

### 2.4. Zika Virus Replication in PC3 Cells

#### 2.4.1. ZIKV Growth Curve

The PC3 and Vero E6 cells were infected with a MOI= 1 ZIKV COL345Si isolate for 2 h (37 °C at 5% CO_2_). Next, the wells were washed with PBS, and DMEM 2% FBS was added 2, 6, 12, 24, 48, 72, 96, and 120 h post-infection (h.p.i.). The supernatants and monolayers cells were collected for virus titration by plaque-forming units and to determine the amount of the positive and negative viral RNA genome, respectively.

The Vero E6 cells were infected with supernatants from PC3, and Vero E6 was infected with ZIKV for 2 h at 37 °C (5% CO_2_), and the wells were washed with PBS and late, 1.5% Carboxymethyl cellulose (CMC), and DMEM supplied with 2% SFB medium solution was added. The Vero E6 cells were included as a control and were treated as described before in [17].

#### 2.4.2. Viral RNA Quantification

The viral RNA extraction was performed with Qiagen’s RNeasy commercial kit (NC9307831, Hilden, Germany) on the infected cells, and we always conserved an environment that was RNAse-free. For the quantification of the positive RNA viral copies, the synthesis of the cDNA of ZIKV was performed using the SuperScript™ III Reverse Transcriptase kit of Thermo Fisher Science, US (18080051, Waltham, MA, USA) and a random hexamer primer of Promega (PR-C1181, Madison, WI, USA). The viral copy number was estimated by qPCR using ZIKV-specific primers, forward: 5′ CTGTGGCATG AACCCAATAG 3′, and reverse: 5′ ATCCCA-TAGAGCACCACTCC 3′, and SsoFastTM EvaGreen ^®^Supermix BIORAD (172 5204, Hercules, CA, USA) [18]. Moreover, the quantification of the negative RNA viral copies was carried out through the synthesis of the cDNA antigenome of ZIKV which was performed using SuperScript™ III Reverse Transcriptase kit of Thermo Fisher Science, US and a forward primer, 5′ CAATATGCTG AAACGCGAGAGAAA. Subsequently, the cDNA synthesis and the viral copy number were estimated as above mentioned [18]. The viral RNA content was normalized to the housekeeping gene GAPDH (ΔΔCt), and it is reported as the fold-change.

#### 2.4.3. Cell Viability Assay

In order to establish the percentage of cell viability in the infected cells, the 3-(4,5-dimethylthiazol-2-yl)-2,5-diphenyltetrazolium (MTT) technique was used. The PC3 and Vero E6 cells were infected at MOI = 1. The cell viability was determined at 6, 12, 24, 48, 72, 96, and 120 h.p.i. according to Roa-Linares et al. 2016 [19]. Finally, the absorbance reading was performed in an ELISA reader (Microplate reader, BioRad) at a wavelength of 570 nm (OD570). The absorbance data were analyzed in the statistical program GraphPad Prism 5.0, and they are expressed as the mean ± standard deviation (M ± SD) of two independent experiments which were performed in triplicate [19].

### 2.5. Evaluation of Antiviral Response in PC3 Cells

The total RNA was extracted with the Qiagen RNeasy commercial kit following the manufacturer’s instructions. For the cDNA synthesis, the RevertAid Minus First Strand cDNA Synthesis Kit (Thermo Fisher Scientific) was used according to the manufacturer’s instructions. The primers used to quantify the mRNAs have been previously reported [20]. Furthermore, the cDNA products were amplified by RT-qPCR using a set of primers specific for various genes as described in Table 1.

The relative expression of each target gene was normalized to the uninfected control and to the housekeeping gene GAPDH (ΔΔCt), and they are reported as the fold-change. Fold-changes of 0.5 and 1.5 were considered as the down- and up-regulation of the gene expression, respectively.

## 3. Results

### 3.1. The Colombian Zika Virus Isolate COL345Si (MH179341.1) Has Characteristics Similar to the Asian-American Lineage

In order to know more about the phylogenetic history of the Colombian isolate in comparison with other available full-genome sequences from GenBank, which have distinct geographic origins (Puerto Rico, French Polynesia, Cambodia, Uganda, and Dakar), we used the New Guinea Dengue Virus (DENV) strain as the outgroup for this inference; DENV is a member of flavivirus family, but it is phylogenetically more distant.

After the respective alignment of the sequences, we confirmed the Colombian isolate, and similar to the other contemporary strains, it retains mutations in the protein NS1: A188V, prM: S139N, the envelope protein: S154N, and NS5: M2634V (Figure 1). Other mutations in the structural and non-structural proteins are also reported [21] (Appendix A). It should be noted that the Colombian isolate when it is compared to the other strains, has a mutation in the capsid, a substitution of aspartic acid to glutamic acid at position 107 (C: D107E) (Appendix A).

The phylogenetic inference showed that the two Colombian isolates (from the Meta and Sucre provinces) are more closely related to the Puerto Rican Zika viruses, although with a modest but important consistency index. These strains, together with the isolates from French Polynesia and Cambodia, constitute one clade (the Asian lineage), which are evolutionarily more distant to the African strains, which is another clade that is well recognized in base of the consistency index.

### 3.2. AXL and TIM-1 Receptors Were Detected on Prostate Adenocarcinoma Cells (PC3)

Taking into account that AXL and TIM-1 are receptors for several flaviviruses and even the Ebola virus [22], we evaluated the expression of the AXL and TIM-1 receptors, and these were consistent in all of the methodologies that we used (Appendix A, In-Cell Western results).

AXL in the PC3 (Figure 2a,b) cells and the VeroE6 (Figure 2e,f) cells showed a granular expression pattern with an important location at the cell membrane (indicated by arrows). Likewise, the TIM-1 expression was observed in the cell lines that are mentioned above (Figure 2c,d,g,h) (see Appendix A to see the image amplified).

After the examination of the subcellular distribution of the receptors for ZIK, with the intention of determining the impact of the viral infection on the expression of the receptors, the cells were inoculated at a MOI = 1 with ZIKV during 24 h.p.i., but the receptors expressions were not altered (Figure 3). However, when the PC3 cell line was infected at a MOI = 3, the AXL receptor expression increased (Appendix A). 

### 3.3. The Colombian ZIKV Isolate Replicates in PC3 Cells

Once the expression of the cellular receptors for ZIKV were seen, the role of the viral growth dynamics was evaluated. To determine the percentage of the infection of the ZIKV, the PC3 and Vero E6 cells were inoculated at several MOIs (0.1, 1, and 3) (Appendix A). Considering the results, MOI = 1 was chosen for infecting the cells and to evaluate the virus replication cycle. Viral kinetics using the plaque-forming units in the PC3 (Figure 4a) and Vero E6 (Figure 4c) cells showed that viral eclipses occur at 6 and 12 h.p.i. The release of virions was observed at 24 h.p.i., reaching an elevated peak at 72 h.p.i. The PC3 cells (Figure 4a) perform constant virus production up to 120 h.p.i, without alterations in the cellular metabolic activity. In contrast, in the Vero E6 cells, the virion release decreases at 120 h.p.i (Figure 4c), and this finding is consistent with loss of cell viability.

Although the production of the virions in the PC3 cells is constant, the synthesis of the viral genomes was detected. Specifically, the positive and negative-stranded RNA genomes show an increase at 24, 48, and 72 h.p.i (Figure 4b), with decreases at 96 h.p.i. (Figure 4b). This behavior was not detected in the Vero E6 cells, where the genome synthesis was constant and decreased.

### 3.4. ZIKV Modulates the Antiviral Response on PC3 Cells

Taking into consideration the dynamics of the viral genomes synthesized in the PC3 cells, the infection times of 6, 72, 96, and 120 h.p.i. were chosen to assess the antiviral response. Six h.p.i was evaluated to determine the response during the early infection stage.

The antiviral response profile showed the presence of an inhibitor of the antiviral response, SOCS-1, from 6 h.p.i to 120 h.p.i. (Figure 5a). The expression of the antiviral genes (ISGs) induced by interferon-β such as OAS-1, Viperin, and SOCS-1 were detected at 6–120 h.p.i. (Figure 5a). OAS-1 and Viperin are interferon-inducible proteins that regulate some of the steps of the viral replicative cycle, while in contrast, SOCS-1 inhibits the antiviral response. Furthermore, the Vero E6 cells did not show an immune response in contrast to the PC3 cells. Therefore, the ISGs were not detected (Figure 5b).

## 4. Discussion

Male fertility demands the cooperation of the several organs of the urogenital system, each carrying out their own function [7]. Through their interactions, the testes, the epididymis, and the male accessory glands such as the prostate contribute to the production of the human seminal plasma [7]. Some reports show that the male reproductive system is a target for viral infections, such as ZIKV. Alexander G. Pletnev et al. in 2021 demonstrated in AG129 mice (with immunodeficiency) that the epithelium of the epididymis transmitted ZIKV, and it came from the testicles, in the early stage of the infection, approximately at 10 d.p.i, and that the accessory gland cells are involved in a complementary viral transmission because they have lower replicative yield compared to the epididymis and testes [23]. The accessory glands participate in the secretion of proteins, growth factors, and other components; each element is necessary for the viability of the sperm and its biological processes [7]. ZIKV RNA was detected in the semen of three vasectomized men up to 69 days after the infection began [4,24], suggesting that ZIKV replication must be performed in urogenital tissues other than the testes [25] because, considering that the ductus deferens are transected during a vasectomy [26], there should not be evidence of ZIKV infection from the testes in the ejaculate of a vasectomized man. For this reason, the prostate gland is an organ of interest in the study of Zika virus infection. In this work, PC3 cells were chosen despite being a tumor line because they have a functional interferon signaling pathway [14]. This allowed us to know the relationship between the antiviral response and the replication of the Zika virus in these cells of up to 120 h.p.i.

Since the Colombian isolate conserves several of the mutations from the pre-epidemics strain of Asian lineage MH179341.1, such as NS1: A188V, prM: S139N, and the envelope protein (E): S154N and NS5: M2634V (Figure 1) [21], it is expected that this phylogenetic history contributes to several aspects of the viral biology of ZIKV that have been investigated here.

Among these phenotypic behaviors that are ascribable to those mutations, there is a remarkable point to be made that is closely linked to the persistence of ZIKV in MRT such as the NS1 protein which decreases the phosphorylation of TBK1, and IRF3 and suppress the induction of IFN-β [27]. In contrast, the substitutions in the PrM [28] and E [29] proteins are related to neurovirulence, and apparently, the NS5 mutation does not alter the viral replication [30]. Additionally, in this study, a mutation was found in the capsid protein mutation, C: D107E (Appendix A), in the MH179341.1 strain, however, this substitution is not relevant, nor does it affect the virus assembly as the amino acids that have been replaced have similar properties [31]. Evidence for this is the efficient production of the viral particles at the evaluated times.

The expressions of the AXL and TIM-1 receptors were demonstrated in the PC3 and Vero E6 cells, and thus, they were defined as being susceptible to a ZIKV infection (Figure 2). It was verified through viral kinetics, which was evaluated at 24–120 h.p.i when we conducted the study, that the cycle of viral replication ends at 24 h, and this was evidenced by the RNA viral genome quantification and PFU production. Hence, the tested cells are permissive (Figure 4). In contrast with Vero E6, the Zika virus infection in the PC3 cells did not alter its viability (Appendix A) Additionally, the results showed the synthesis of genomes with the production of the viral particles at 24–120 h.p.i. (Figure 4). This finding led us to evaluate the antiviral profile at several times (6, 72, 96, and 120 h.p.i), where the highest/lowest yields of the viral genome were evidenced (Figure 5). The Zika virus infection activated the interferon pathway early, as it was evidenced by the detection of interferon-β mRNA and their inducible genes (OAS-1, viperin). It should be noted that SOCS-1, a suppressor molecule of the interferon pathway, is expressed starting from 6 h.p.i, this means that there is an early negative regulation. Other research shows the variability profile of the IFNs (IFNα or IFNγ) and IFN-related proteins in the PNT1A cells (normal prostate epithelial) and the 19I cells (stromal MSCs, derived from a healthy prostate donor) infected with several strains of ZIKV, and the authors of these studies suggest that these differences should be modifications and phenotypic changes in the virus by the passages on human cells and insects cells (C636) [32]. The reports indicate that viral passages at 10–100 can affect the pathogenicity and infectivity of the virus and its sensitivity to antiviral treatments [32,33]. In our study, we used a ZIKV isolate of *A. aegypti* at passage #6 (COL345Si), and by comparing it with other contemporary strains (human isolate in Colombia-Meta MH544701 and the PRVABC59 strain, Puerto Rico Isolated), we found minimal genetic variations; the COL345Si strain has a mutation in the capsid protein that is mentioned above (Appendix A). In fact, the viral yield of the PRVABC59 strain in the HUVEC cells is similar to the viral kinetics that are found with the ZIKV-COL345Si in the PC3 cells, and both of the viral strains activate the interferon type I response. It is relevant to mention that the antiviral response depends on the virus–cell interaction type. For example, in Hofbauer cells infected with ZIKV- PRVABC59 [34], the antiviral response reduces the viral replication, but in our case, we observed that this response is not sufficient to clear the ZIKV infection in the PC3 cells. The presence of the AXL receptor in the PC3 cells could explain the behavior in the immune response.

It has been described that when AXL binds to the interferon receptor, they activate and phosphorylate Janus kinase 1 (JAK1); the latter one phosphorylates the STAT1 transcription factor which translocates to the nucleus and promotes SOCS-1 transcription [12]. Recent reports have identified AXL as a potential factor of ZIKV entry and a possible antiviral response regulator. When the kinase action of AXL is inhibited, a decreased rate of entry of the ZIKV virus was not detected, however, alterations were observed in the expression of SOCS 1 and 3 and of the interferon-stimulated genes, and there was a decrease in the replication rate of the ZIKV virus [13]. These data demonstrates that AXL promotes ZIKV entry and negatively regulates the antiviral state of the Sertoli cells to augment the ZIKV infection of the testes, and this provides new insights into testis antiviral immunity and possible ZIKV persistence [13]. Other research has demonstrated that the participation of SOCS-1, which modulates the Zika virus lineages’ replication in different cell lines A549, JAr, and hNPC, will be reduced [35]. This participation was confirmed by the fold-change of the SOCS-1 and SOCS3 mRNAs in infected cells [35]. Additionally, the protein expression of SOCS 1 and 3 was evaluated by a Western blot [35], and as in our study, the expression of these genes was detected in the first hours of the infection. Thus, it was demonstrated that mRNA expression is correlated with the expression of the protein [35].

Moreover, the viral genome synthesis has a constant production of up to 72 h.p.i, and it diminishes at 96 h.p.i. These findings suggest that interferon response and SOCS-1 could modulate the viral replication. This concurs with other research, demonstrating that prostate cells (LNCaP) and human prostate mesenchymal stem cells (MSCs) infected with ZIKV do not show differences in replication due to the absence of the interferon response [36]. On other hand, MSCs cells perform functional interferon signaling, and therefore, there is ZIKV strain replication variability [36].

Consistent with other studies that were conducted in Zika-infected Human Brain Microvascular Endothelial Cells (hBMECs), the ISGs were detected up to 9 days after the infection began (d.p.i.), however, there were variations in the profiles of the type I interferons of up to 2 d.p.i. Apparently the mechanisms of regulation of the antiviral response are different to those that were observed in this research [37]. In addition, the lack of a cytopathic effect could be due to the induction of the pro-survival genes during the ZIKV infection of hBMECs, including EGR1, ATF3, and BIRC3 [37]. In our study, we did not evaluate the survival molecules, but it cannot be ruled out that these are responsible for the absence of a cytopathic effect at up to 120 hpi.

We propose that AXL exerts control over the antiviral response in a ZIKV infection, as when cells are infected at a high MOI (MOI = 3), the receptor expression increases, unlike in TIM-1, so the latter only has a role in virus entry. Therefore, allowing us to raise a possible cross-talk between AXL and type I interferon signaling is essential for modulating the antiviral state and the replication of ZIKV at 120 h.p.i.

It is important to mention that the African and Asian strains have differences in their in vitro replication rates among the mammalian and insect cell lines [2,38]. For example, the African lineage replicates faster, unlike the Asian lineage, causing cell death [34]. Due to this, our results contrast the findings that were observed with the Colombian Zika Virus isolate COL345Si (Asian-American lineage) with ZIKA Dakar (African lineage) infected PC3 cells, and this study revealed that ZIKA-Dakar caused a cytopathic effect (CPE) in the PC3 cells from 5 to 10 dpi (Appendix A). These results are similar to other findings that demonstrated that differential cell lines are susceptible to ZIKV Uganda (African) and ZIKV PRVABC59 (Asian-American) infections, where the authors found the LNCaP (prostate) and 833KE (testis) cells lines were permissive to viral infection without a CPE after the infection. Therefore, the differences between the viral kinetics of the Vero E6 and PC3 cells and the additional results that were obtained with the strain Dakar allow us to hypothesize the possible persistence of the ZIKV-COL345Si strain infection in prostate cells, nonetheless, it is necessary to perform in vitro culture tests over prolonged infection times to identify the regulation of the cellular genes involved in this phenomenon.

Taking the data about the distinct variables of the biology of the ZIKV Colombian isolate and its potential phylogenetic history, some reflections can be made. The origin of ZIKV in these assays is compatible with the Asian lineage in terms of some of the genetic variants, but after the epidemic break in 2014, the evolutionary landscape changed. Surely, many factors contributed to the shaping of the Colombian variant of ZIKV, including the highest rates of viral transmission in a new habitat that was previously unknown for the virus in America. This investigation shows an adaptive process in which the viruses have transformed in a persistent way in the male reproductive tract. How those ancestral genetic variants of African ZIKV have survived in those sexually transmitted cases in humans, changing to a schema of persistent infection, is a question still without definitive answers. The perspective of this research could be focused on the future experimental evolution studies in vitro for modeling, in a more complete way, the phenomenon of viral persistence and how some viruses according to ecological and evolutionary opportunities can change tropism and other important characteristics which are distinctive of the viral biology of each virus.

## Figures and Tables

**Figure 1 microorganisms-10-02420-f001:**
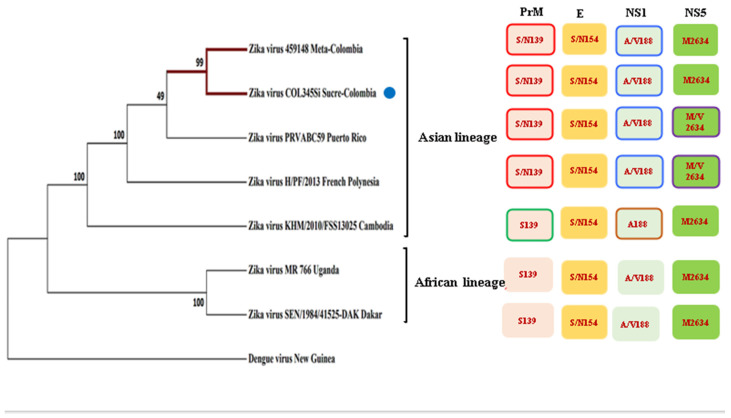
Phylogenetic tree of several strains of the Zika virus in comparison with the Zika isolated COL354Si GenBank: MH179341.1 (indicated with a blue point) showed similar mutations to the other Asiatic strains.

**Figure 2 microorganisms-10-02420-f002:**
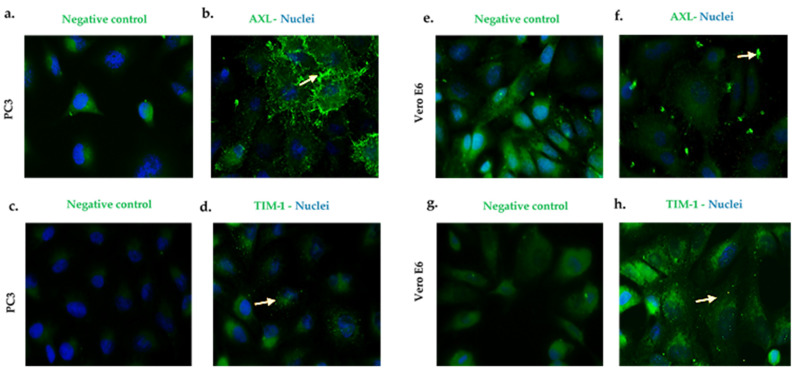
PC3 and Vero E6 cells express AXL and TIM-1 cell receptors. Pictures were captured at a 60× magnification. The photographs represent the subcellular location of AXL (**b**,**f**) and TIM-1 (**d**,**h**), and they correspond to three independent triplicate experiments. AXL and TIM-1 (green), and nuclei (blue). (**a**,**c**,**e**,**g**) are cells antibody secondary labeled and correspond to the negative controls.

**Figure 3 microorganisms-10-02420-f003:**
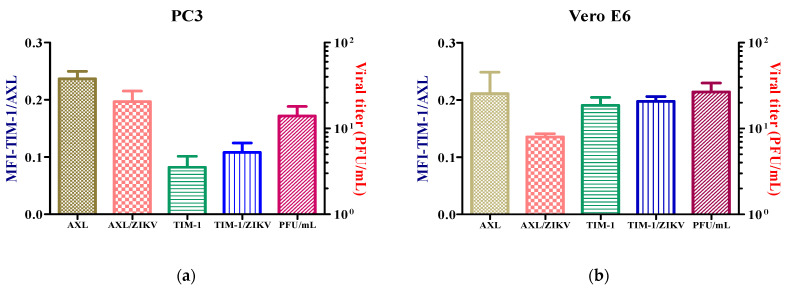
Expression of AXL and TIM-1 receptors in PC3 (**a**) and Vero E6. (**b**) The infected and uninfected cells after 24 h were detected by flow cytometry. Analysis shows mean and SEM (Standard Error Medium) from independent experiments.

**Figure 4 microorganisms-10-02420-f004:**
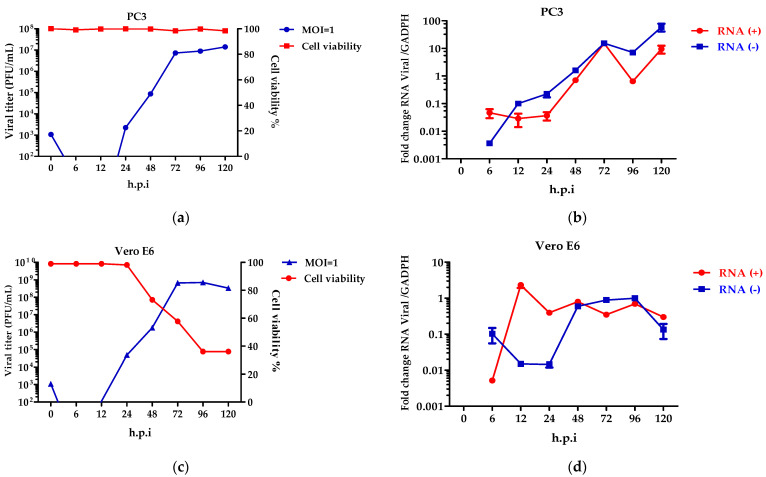
ZIKV replicates in PC3 cells. Viral kinetics was determined in PC3 (**a**,**b**) and Vero E6 (**c**,**d**) cells infected at an MOI = 1 at 0–120 h.p.i, and 0 h.p.i corresponds to viral input. Characterization of the replicative cycle was established by plaque-forming units (**a**,**c**) and quantification of viral genomes (**b**,**d**). Analysis shows mean and SEM from independent experiments.

**Figure 5 microorganisms-10-02420-f005:**
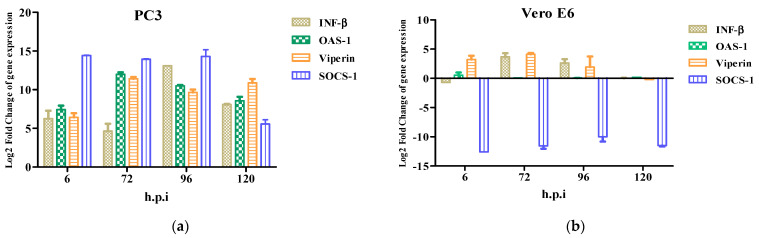
Innate immune response to Zika virus in PC3 (**a**) and Vero E6 (**b**) cells were evaluated at 6, 72, 96, and 120 h.p.i by RT-qPCR. mRNA expression of β and antiviral proteins (OAS1, Viperin, and SOCS-1) in ZIKV-infected PC3 (**a**) and Vero E6 (**b**) were determined. Data are presented as the mean ± SEM from three independent experiments.

**Table 1 microorganisms-10-02420-t001:** The interferon- β and genes antivirals primers.

Gene	*Forward*	*Reverse*
INF-β1	5′-CGCCGCATTGACCATCTA-3′	5′-GACATTAGCCAGGAGGTTTCTCA-3′
Viperine	5′-AAATGCGGCTTCTGTTTCCAC-3′	5′-TTGATCTTCTCCATACCAGCTTCC-3′
Oligoadenylate synthetase I (OAS-1)	5′-GTGTGTCCAAGGTGGTAAAGG-3′	5′-CTGCTCAAACTTCACGGAA-3′
Suppressor of cytokine signaling 1 (SOCS-1)	5′-CACTTCCGCACATTCCGTTC-3′	5′-CACGCTAAGGGCGAAAAAGC-3′
GAPDH	5′-AGCCACATCGCTCAGACAC-3′	5′ GCCCAATACGACCAAATCC-3′.

## Data Availability

No applicable.

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
