# Peer review of "The Colombian Zika Virus Isolate (COL345Si) Replicates in Prostate Adenocarcinoma Cells and Modulates the Antiviral Response"

_microorganisms, 2022, doi:10.3390/microorganisms10122420_

Round 1

Reviewer 1 Report

In this manuscript Yaneth M-B and co-authors examined the ability of Zika virus (ZIKV) to propagate and elicit antiviral response in human prostate cells using PC3 adenocarcinoma cells as a model. Authors demonstrated that PC3 cells express ZIKV entry receptors AXL and TIM1They also showed accumulation of viral RNA and production of infectious particles by ZIKV-infected PC3 cells. They also show that ZIKV infection induces expression of ISGs in PC3, but not Vero cells. Authors conclude that ZIKV can infect prostate cells, which can contribute to sexual transmission of the virus. Authors also make several other curious observations, but in my opinion they have to be verified. My major concers are the following:

1.     Materials and methods are described very briefly and don’t provide sufficient details to allow reproducibility of the techniques. In particular it applies to in-cell western, qRT-PCR and plaque-assay.

2.     Figure 2 requires negative control with cells that don’t express AXL and TIM1. The negative control shown in the figure is not explained.

3.     What are the error bars presented on all graphs? This needs to be specified in figure legends?

4.     Figure 4 b shows sudden drop and then rise of viral RNA level in PC3 cells at 48hpi. This is very strange and has not been observed in any other cells. Therefore it is important to know how many times this experiment was repeated showing the same results. Given that this could have resulted simply from RNA degradation, authors should present the evidence for RNA integrity in the samples used for this experiment. Authors should also include a positive control (expression of cellular gene) to show that PCR with this samples was efficient.

5.     In figure 5 authors show induction of ISGs by ZIKV infection in PC3, but nor VeroE6. This was done by PCR using the same set of primers for human (PC3) and monkey (Vero cells). Did authors check that whether these primers efficiently recognise monkey genes?

6.     Statement in line 298-301 is not supported by the evidence and needs to be removed.

Author Response

Dear reviewer

Thanks for the recommendations

Next, attach each one

Regards

Reviewer 2 Report

In this study, the authors found that PC3 prostate cell line is AXL and TIM-1 positive, making it susceptible and permissive to ZIKV infection. Interestingly, ZIKV infection of PC3 cells did not alter viability of PC3 cells or cause cytopathic effect. The mRNA level changes of several antiviral response related genes were detected by RT-qPCR. Overall, this study is interesting, but control maybe inappropriate and the evidences are not enough for convincing. Below are concerns and suggestions to improve the manuscript before further consideration.

1. In Line 59 – 61, authors emphasized that the immortalized prostate epithelial cell (PC3) was used because this cell line has the functional pathway of the interferon response. However, the Vero E6 cell line, which was used as control for comparison in this study, is of genetic defect in interferon signaling pathway (Please see the references below). Cell line with functional interferon pathway should added as control. The results from Vero E6 cells could also be displayed together.

Some references showing that Vero E6 is defective in interferon signaling pathway:

1) Regulation of the interferon system: evidence that Vero cells have a genetic defect in interferon production. J Gen Virol. 1979 Apr;43(1):247-52. doi: 10.1099/0022-1317-43-1-247.

2) New World hantaviruses activate IFNλ production in type I IFN-deficient Vero E6 cells. PLoS One. 2010 Jun 17;5(6):e11159. doi: 10.1371/journal.pone.0011159.

2. One of the main topics of this study is that ZIKV could infect PC3 cells without altering cell viability or causing a cytopathic effect. This unusual phenomenon is very interesting and enough evidences are needed for confirmation. However, only the result of MTT experiment is shown, which is not sufficient to draw a conclusion. Even the picture showing the CPE of ZIKV Dakar strain infected CP3 cells was displayed in supplementary materials, there is no picture of Colombian isolate of ZIKV COL345Si infected CP3 and Vero E6 (and other IFN signal pathway competent cell line). So, at least, in Fig. 4, the picture of PC3 cells and picture of Vero E6 (and other cell line with functional IFN signaling pathway) infected with ZIKV COL345Si (MH179341.1) at 120 hpi should be shown, together with the picture of  mock infected cells.

3. In Fig. 2, the green signals could also be seen in all of the Negative control. What does the negative control mean? In Fig. 2g (negative control), at least one green granular spot is very obvious. However, in Fig. 2h, green granular spot can hardly be seen where the arrow indicated. Something wrong with this figure?

4. In Fig. 3, the original flow cytometry histograms should be displayed like the Fig. 4 in the paper ‘Replication of Zika Virus in Human Prostate Cells: A Potential Source of Sexually Transmitted Virus’.

It seems that it is the ratio of MFI that was shown. So, what was used for normalization?

5. In Fig. 3, the viral titer is around 20 PFU/ml. Why is the titer so low?

6. In Line 96-99, the authors described that at 2, 6, 12, 24, 48, 72, 96, and 120 hpi, supernatants and monolayers were collected for virus titration by PFU and to determine the amount of positive and negative viral RNA genome. However, in Fig 4, all of the data points are collected at 0, 6, 12, 24, 48, 72, 96, and 120 hpi. There is no data point at 0 hpi for ZIKV RNA(-) level in Fig. 4d. There is no data point at 6 and 12 hpi for viral titer in Fig. 4a and in Fig. 4c.

In Fig. 4a and 4c, the viral titers are about 10^3 PFU/ml at 0 hpi. Is the titer at 0 hpi the original concentration of ZIKV for infection?

7. In Fig. 4b and in Fig. 4d, authors used ratio to show the viral RNA level. Was the level of GAPDH used for normalization? And it is very interesting that the level of ZIKV RNA (+) and RNA (-) sharply descended from 24 hpi to 48 hpi and sharply ascended from 48 hpi to 72 hpi, then descended from 72 hpi to 96 hpi again. How many times was this experiment repeated, and did the author get the similar results? How about the repeatability. This should be described in the manuscript.

8. It is the proteins that exert the antiviral activity or inhibit the antiviral response, not the mRNA of the gene. And sometimes the mRNA levels of genes are not proportionate to the protein levels. So, besides fold change of mRNA by RT-qPCR, the protein level of INF-β, OAS-1, Viperin and SOCS-1 inside cells should be quantified by traditional western blot. The secreted INF-β can be quantified by ELISA kit. In principle, the protein levels should be determined by western blot at the same time point for viral RNA to try to explain the fluctuations detected in the viral genome synthesis. For simplification, western blot at 0, 24, 48, 72 and 96 hpi should be performed.

9. Although there is fluctuation in viral genome synthesis inside cells, the released virions rapidly and steadily increased from 24 hpi to 72 hpi without fluctuation and almost reached a plateau from 72 hpi to 120hpi. What is the function of viral genome synthesis fluctuation in persistent ZIKV infection of PC3 cells?

Minor concerns:

1. Line 98, How was the plaque-forming units experiment performed? It should be descried in the Material and Methods section.

2. Line 89 -92, there is only catalogue number for anti-AXL antibody and Hoechst33342 and there is only company name for anti-TIM-1 antibody, Qiagen's RNeasy commercial kit (Line 102) and SuperScript™ III 103 Reverse Transcriptase kit (Line 104). There is no information for Alexa Fluor 488 secondary antibody. Both company name and catalogue number should be given.

3. In Fig. 1, GenBank accession number of each strain should be written after the strain’s name.

4. The legend of Fig. 5 is incompletely displayed.

Author Response

(The authors gave the same response as above.)

Reviewer 3 Report

This study reports that ZIKV replicates in PC3 prostate cell line without altering cell viability or causing a cytopathic effect and modulate the antiviral response through the suppressor molecule expression, SOCS-1. I have several concerns regarding the paper

1.     There is no information about the Colombian isolate of the Zika Virus COL345Si particularly how it was isolated and from whom and the demographic and disease details of the patient. What was the passage number of virus. The authors are requested to refer Clinton et al 2020 where it has been mentioned that passage history affects virus replication in prostate cell lines

2.     The methodlogy and results are not clear and easy to understand. Particularly the figure legends need to be more informative

3.     The PFU/ml is 101 in Figure 3 while it is 107-11 in Figure 4 why such huge difference.

4.     In figure 4, Viral RNA levels has been mentioned in a scale of 0 to 1. What does it mean copy number or what?

5.     Figure 2 is not clear and even in negative control, it is looking green.

6.     How the fold change in RNA levels were calculated? Are these values significantly different.

7.     The downstream effect of SOCS upregulation has not been studied, How authors suggest that ZIKV modulate antiviral response through SOCS1

8.     In Supplementary figure 5, ZIKV-Dakar show CPE in PC3 while the same images for Colombian isolate have not been provided which will provide a conclusive evidence that Colombian isolate does not form CPE

9.     No information about supplementary image 4 in manuscript?

10. If ZIKV -dakar forms CPE and Colombian isolate does not form CPE, is there any known genetic differences

11. Line 262: cycle of viral replication ends at 24 hours- results and discussion differs

12. The manuscript has many grammatical errors. In methods, future tense used in some places.

13.  Following references  related to ZIKV in prostate cells needs to be included and discussed

14. How Colombian isolate differs from others to be mentioned in title?

Pletnev AG, Maximova OA, Liu G, Kenney H, Nagata BM, Zagorodnyaya T, Moore I,

Chumakov K, Tsetsarkin KA. Epididymal epithelium propels early sexual

transmission of Zika virus in the absence of interferon signaling. Nat Commun.

2021 Apr 29;12(1):2469. doi: 10.1038/s41467-021-22729-5. PMID: 33927207; PMCID:

PMC8084954.

Spencer Clinton JL, Tran LL, Vogt MB, Rowley DR, Kimata JT, Rico-Hesse R.

IP-10 and CXCR3 signaling inhibit Zika virus replication in human prostate

cells. PLoS One. 2020 Dec 30;15(12):e0244587. doi: 10.1371/journal.pone.0244587.

PMID: 33378361; PMCID: PMC7773246.

Delafiori J, Lima EO, Dabaja MZ, Dias-Audibert FL, de Oliveira DN, Melo CFOR,

Morishita KN, Sales GM, Ruiz ALTG, da Silva GG, Lancellotti M, Catharino RR.

Molecular signatures associated with prostate cancer cell line (PC-3) exposure

to inactivated Zika virus. Sci Rep. 2019 Oct 25;9(1):15351. doi:

10.1038/s41598-019-51954-8. PMID: 31653965; PMCID: PMC6814752.

Chan JF, Yip CC, Tsang JO, Tee KM, Cai JP, Chik KK, Zhu Z, Chan CC, Choi GK,

Sridhar S, Zhang AJ, Lu G, Chiu K, Lo AC, Tsao SW, Kok KH, Jin DY, Chan KH, Yuen

KY. Differential cell line susceptibility to the emerging Zika virus:

implications for disease pathogenesis, non-vector-borne human transmission and

animal reservoirs. Emerg Microbes Infect. 2016 Aug 24;5(8):e93. doi:

10.1038/emi.2016.99. PMID: 27553173; PMCID: PMC5034105.

Minor mistakes

Abstract: Zika virus (ZIKV), a flavivirus mainly transmitted by A. aegypti and A. albopictus, sexual transmission  ------------mosquito names to be in italics. There is no link between part of sentence after comma.

Line 32 The persistence of viral RNA in different MRTs, like C57BL/6J mice……………… viral RNA in MRTs of C57BL/6J

Author Response

(The authors gave the same response as above.)

Round 2

Reviewer 2 Report

Thanks for the comments. Below are the concerns unresolved.

1. Usually, the negative control that was only labeled with secondary antibody should show no signals. There are obvious green signals in Fig. 2a, 2c, 2e, 2g. Why?

Indeed, the condensed granular expression pattern at the cell membrane could be seen in Fig. 2b and 2f compared with 2a and 2e. However, the granular expression pattern could NOT be observed in Fig. 2d and 2h. The green signals in the cytoplasm of Fig. 2d are slightly stronger than that of Fig. 2c.  The green signals in the cytoplasm of Fig. 2h are almost the same as that of Fig. 2g. So, it may be incorrect to say that the TIM-1 expression was observed.

2. In the knock-out mice, the genes are removed and the expression of the corresponding proteins disappear. So, the research by using gene knock-out mice did not support the idea that the result of mRNA level could substitute for the result of protein expression level.

In the Abstract, one of the main conclusions is that during ZIKV infection the PC3 cells modulate the antiviral response through the SOCS-1 expression. The result of mRNA level CANNOT substitute for the result of protein expression level. So, to see how PC3 cells modulate the antiviral response through the SOCS-1, at least the western blot of SOCS-1 in infected PC3 and Vero E6 should be done with uninfected cell as control.

3. In the new Fig. 4b of revised manuscript, both the positive and negative strand of ZIKV RNA rise steadily, while in the previous Fig.4b the level of RNA (+) and RNA (-) sharply descended from 24 hpi to 48 hpi and sharply ascended from 48 hpi to 72 hpi, then descended from 72 hpi to 96 hpi again.

The slight drop at 96 hpi in new Fig.4b is not a violent fluctuation which was seen in the previous Fig.4b. And the slight drop is probably fortuitous phenomenon.

So, if the new Fig.4b is correct, the description of the fluctuation in the manuscript should be removed.

Minor concerns:

1. In the legend of Fig.4, it should be described that the viral titer at 0 hpi is the input.

2. Line 124, What does CMC stand for?

Author Response

Dear Reviewer, 

Thanks for the recommendations were so interesting and support our work

Kind regards

Reviewer 3 Report

The authors have addressed the comments but still many point needs clarification.

Figures in the revised manuscript are not clear since the deleted and new figure overlaps each other.

Line 84 : the infection was incubated…………….modify to “infected cells were incubated”

In supplementary figure 5, the bars for Zika Dakar is missing

In fig 4, the PFU titer is increasing constantly but the same for RNA is not why. The authors need to present the raw data for RNA levels without normalizing also. The methodology for normalization need to be provided.

The authors have studied only upto 5 days for PC3. Since the virus PFU has not decreased, can they extend the study for more days to see what happens after 5 days. It is possible that at later stage the PC3 cells might show apoptosis or CPE?

Author Response

Dear reviewer

Thanks for the recommendations were so interesting and support our work

Kinds regards
